# The Structure of Maltooctaose-Bound *Escherichia coli* Branching Enzyme Suggests a Mechanism for Donor Chain Specificity

**DOI:** 10.3390/molecules28114377

**Published:** 2023-05-27

**Authors:** Remie Fawaz, Courtney Bingham, Hadi Nayebi, Janice Chiou, Lindsey Gilbert, Sung Hoon Park, James H. Geiger

**Affiliations:** 1Department of Chemistry, Michigan State University, East Lansing, MI 48824, USA; rfawaz@itu.edu (R.F.); bingha23@msu.edu (C.B.); hadi.nayebi@dandeliontx.com (H.N.);; 2Department of Food Service Management and Nutrition, College of Natural Sciences, Sangmyung University, Hongjidong, Jongnogu, Seoul 03016, Republic of Korea; parksh@spc.co.kr

**Keywords:** branching enzyme, glycogen biosynthesis, starch, glycosyl hydrolase, GH13

## Abstract

Glycogen is the primary storage polysaccharide in bacteria and animals. It is a glucose polymer linked by α-1,4 glucose linkages and branched via α-1,6-linkages, with the latter reaction catalyzed by branching enzymes. Both the length and dispensation of these branches are critical in defining the structure, density, and relative bioavailability of the storage polysaccharide. Key to this is the specificity of branching enzymes because they define branch length. Herein, we report the crystal structure of the maltooctaose-bound branching enzyme from the enterobacteria *E. coli*. The structure identifies three new malto-oligosaccharide binding sites and confirms oligosaccharide binding in seven others, bringing the total number of oligosaccharide binding sites to twelve. In addition, the structure shows distinctly different binding in previously identified site I, with a substantially longer glucan chain ordered in the binding site. Using the donor oligosaccharide chain-bound *Cyanothece* branching enzyme structure as a guide, binding site I was identified as the likely binding surface for the extended donor chains that the *E. coli* branching enzyme is known to transfer. Furthermore, the structure suggests that analogous loops in branching enzymes from a diversity of organisms are responsible for branch chain length specificity. Together, these results suggest a possible mechanism for transfer chain specificity involving some of these surface binding sites.

## 1. Introduction

Glycogen and starch are the primary energy storage molecules for all kingdoms of life. Both consist of α-1,4-linked glucose units with sporadic α-1,6 branches [1]. However, the tertiary structures of glycogen and starch are distinct, with glycogen molecules existing as highly branched clusters that have a glycogenin protein molecule at their center. Starch, on the other hand, has a more complex structure that consists of two distinct types of molecules: amylose, which has very few branches (two to four chains per 1000 glucose units [2,3]), and amylopectin, which is substantially more branched (two to four branches per 100 linear glucose units). Together, these molecules make up a starch granule consisting of alternating layers of amorphous and semi-crystalline material [2,3]. These basic structures can differ markedly depending on the organism and, in the case of starch, tissue type. In the case of bacterial glycogen, the density and structure of glycogen particles differ based on the organism [4]. Bacterial strains that commonly, but not exclusively, reside in an animal host, such as *E. coli* and other enterobacteria, tend to have much longer and more frequent branches, leading to less compact and more readily digestible glycogen [5,6]. Organisms that spend a significant portion of their life-cycles in a low-nutrient environment, requiring the organism to substantially slow its metabolism, make glycogen with shorter and less frequent branch points, leading to a glycogen that is much less easily digested, slowing its degradation in sync with the slower metabolism required by the low nutrient environment [5,6,7]. Thus, different organisms, and even different tissues in the same organism, must make distinctly different versions of the same storage polysaccharide. The biosynthesis of this material must be tailored to these requirements.

The biosynthetic pathways of glycogen and starch are very similar and involve three enzymes to make the parent glycan and a variety of enzymes that are involved in tailoring the basic material, with debranching enzymes to remove incorrectly placed branches and kinases that phosphorylate glucose units being the most common activities [1,3,7,8]. The three enzymes common to all storage polysaccharide synthesis are first an NDP-glucose pyrophosphorylase (ADP-glucose pyrophosphorylase in bacteria and plants and UDP-glucose pyrophosphorylase in other eukaryotes) [9,10], which synthesizes an activated NDP-glucose unit from either ATP or UTP and glucose-1-phosphate [10,11]. The second step creates the alpha-1,4-glucose linkages that define the polymer and are catalyzed by glycogen or starch synthase. In the third step, catalyzed by the branching enzyme (BE), an alpha-1,4 linkage is cleaved, and its reducing end is reattached to make α-1,6 branch points on the polymer. The frequency, location, and length of these branches play important roles in determining the structure and, in some cases, the bioavailability of the storage polymer. For example, a strong correlation can be made between the specificity of a bacterial organism’s BE, the structure of its glycogen, the relative digestibility of the glycogen, and the organism’s propensity to live in harsher, less nutrient-rich environments [6,12,13]. A number of studies have shed light on this relationship in a wide variety of bacterial organisms, including *E. coli* [14,15], *Synechocystis* [16], *Agrobacterium tumefaciens* [17], *Rhodococcus* [18], *Vibrio vulnificus* [19,20], and *Galdieria sulphuraria* [21,22]. Branching enzymes have also found use in a number of commercial applications, and branch transfer length is of key importance in these applications as well [23].

Most BEs so far identified are members of the enormous GH13 family of glycosyl hydrolases (as defined by CAZY, http://www.cazy.org/GH13.html (accessed on 23 May 2023)), which include alpha-amylases, isoamylases, pullulanases, and a variety of other enzymes that are active on glucan polymers. BEs that are members of the GH57 family have also been identified in some prokaryotes [24]. BEs from prokaryotes are members of the GH13_9 subfamily, and eukaryotic BEs are members of the GH13_8 subfamily. Thus far, the structures of BEs from *E. coli*, *M. tuberculosis*, rice (BE1), humans, *R. obamensis*, *C. glabrata*, and the cyanobacterium *Cyanothece* have been determined [25,26,27,28,29,30,31,32,33]. All of the structures have in common a carbohydrate binding module 48 (CBM48) domain n-terminal to the canonical (βα)_8_ TIM barrel catalytic domain common to all GH13 enzymes and a c-terminal β-sandwich domain common to a number of GH13 enzymes, including the α-amylases, isoamylases, and a number of others [1,8,10,23,34]. However, the n-terminal domains differ between prokaryotes and eukaryotes, with many prokaryotes having yet another β-sandwich domain, while eukaryotes have a helical domain. The location of this n-terminal domain can vary widely among prokaryotes, while the helical domain of eukaryotes appears to be more structurally similar. A number of studies have focused on understanding the relationship between BE isoforms and starch [5,35,36,37] and glycogen structure.

We have been working for many years to understand the enzymes of glycogen and starch metabolism [38,39,40,41,42]. We have recently focused on understanding the structural details of BE, because the details of its interaction with its polymeric substrate will be essential in understanding its unique specificity. We have focused our attention on BE from *E. coli* (EcBE), an *Enterobacteriaceae* organism that most commonly lives in the nutrient-rich guts of warm-blooded animals. We determined the first structure of a BE, EcBE [25,43,44] and subsequently determined the structures of EcBE bound to α, β, and γ-cyclodextrins [44], maltohexaose, and maltoheptaose [43]. Together, these structures identified seven distinct oligosaccharide binding sites, all located more than 15 Angstroms from the active site and all on the surface of the enzyme. While one of the binding sites (site VII) bound only cyclodextrins, three sites (sites I, II, and VI) bound only linear oligosaccharides. We have further shown that several of these binding sites (sites I, IV, VI, and VII) have significant effects on the activity of the enzyme, as mutation results in significant loss of activity [43,44]. Herein, we report the structure of M8-bound EcBE, which, to our surprise, identified an additional five new surface glucan binding sites, creating a large glucan surface concentrated on one side of the enzyme. Using key information from the recently published [29] structure of the M8-bound *Cyanothece* BE, we can now identify the binding sites most likely to be involved in donor chain binding and those most likely to be responsible for the transfer of the longer glucan chains, which is a hallmark of EcBE. The new picture shows how some of the disparate binding sites on the EcBE surface may work together to confer its activity and gives some clues regarding how the enzyme may orient itself on the growing polymer, which is its substrate.

## 2. Results

### 2.1. Overall Structure and Packing of M8-Bound EcBE

As was the case for all previous malto-oligosaccharide-bound EcBE structures, EcBE with its N-domain removed (the first 113 amino acids of full-length EcBE) was used for crystallization, and soaking these previously grown crystals in high concentrations of M8 was the only path to diffraction-quality M8-bound EcBE crystals. A new EcBE crystal form was used in this case, which has different crystal packing interactions, possibly freeing other regions of the molecule for sugar binding. As was the case for previous structures, there are four unique molecules in the asymmetric unit that provide additional possibilities for sugars to bind without destroying the lattice (Figure 1). As shown, not all M8 binding sites are occupied in each of the four molecules in the asymmetric unit, most likely due to the different crystal packing for each molecule. Table 1 enumerates the occupancy of each binding site by molecule in the asymmetric unit and includes a representative residue for each. The binding site numbering used in previous malto-oligosaccharide-bound EcBE structures was preserved here. As shown in Table 1, five of the seven previously identified sites are bound by M8. However, probably due to the large number of binding sites available on each EcBE molecule, resolution was significantly compromised by M8 soaking, giving a structure of relatively low resolution (3.0 Å). This resolution will not allow precise identification of all protein/M8 interactions. Nonetheless, since the very nature of the experiment results in this loss of resolution, this is the highest resolution picture available of the new binding sites identified here. A composite EcBE structure was created by overlaying all four molecules in the asymmetric unit, extracting the best ordered M8 bound in each of the binding sites, and merging the resultant M8 molecules with a single EcBE polypeptide chain. The result of this manipulation is shown in Figure 2, which shows M8 binding in all of the binding sites identified in the M8-bound EcBE structure. In addition to the five sites previously identified in the M7-bound EcBE structure, five additional sites were identified (sites VIII–XII). 

### 2.2. Sugar Binding in Sites I–V

Binding in the five previously identified sites was relatively similar to that seen in the M7-bound EcBE structure for most of the sites, though more glucose units were seen in most of the five sites (compare the 4, 2, 2, 2, and 2 glucose units seen in sites I–V in the M7-bound EcBE structure to the 7, 2, 1, 4, and 3 glucose units seen in the M8-bound EcBE structure). There are also some minor deviations in the sugar positions in the binding sites; however, there are also similar differences when comparing different molecules in the asymmetric unit of the M8-bound structure. These deviations can be attributed to a combination of the lower resolution of the structure, which introduces increased positional error, as well as possibly lattice interactions between the malto-oligosaccharide and adjoining molecules that could force deviations in the binding site. Nonetheless, the same residues found to interact with the glucans in the M7-bound structure are also making similar interactions in the M8-bound structure.

However, M8 binding in site I is quite different from that seen in the M7-bound structure (Figure 3a). As shown, three glucose units, not seen in the M7-bound structure, wrap around the 247–264 loop (which connects β_1_ and α_1_ of the (α,β)_8_ central catalytic TIM barrel domain), ending at the entrance to site VII, suggesting that an even longer glucan would span sites I and VII (Figure 3b). Numerous additional interactions are seen relative to the M7-bound structure (Figure 3). This entire region (binding sites I, II, and VII and the 247–264 loop) is conserved only in *Enterobacteria*, where it is relatively strongly conserved (Appendix A). However, both the loop and the glucan binding sites disappear in more divergent bacteria, archaebacteria, and eukaryotes (Appendix A).

### 2.3. Binding in Sites VIII–XII

In addition to the five malto-oligosaccharide binding sites previously identified in the cyclodextrin- and M7-bound structures, an additional five binding sites (sites VIII–XII) were identified in the present structure (Figure 2 shows the locations of sites VIII–XII). Site VIII interacts with both the catalytic and c-terminal domains and is closest to site V. It binds to the top of the C-domain, making numerous interactions, mostly with the n-terminus of helix 6′ (the second helix between β5 and β6 of the catalytic domain) and two loops from the c-terminal domain (the loops connecting S1 and S2 and S3 and S4 of the c-terminal domain). Four of the five glucose units observed in the structure make direct interactions with the protein, with most of the interactions involving hydrogen bonds between protein sidechains and hydroxyl groups of the sugars (Figure 4a), although there are three residues (Pro497, Pro559, and Val560) that pack against the malto-oligosaccharide as well. There are no stacking interactions involving aromatic amino acids, as is often seen in sugar–protein binding sites.

The next two new binding sites are located on the “side” of the molecule, with the entrance to the active site at the “top” of the enzyme (Figure 2). Site IX encompasses five glucose units spread over a wide surface of the enzyme, with protein interactions with all five glucose units (Figure 4b). There are no ring stacking interactions between aromatic amino acids and glucose rings seen in most of the binding sites. Instead, hydrogen bonding interactions define most of the interactions (Figure 4b). Site X is located between the large glucans found in sites I and IX, with an electron density for only two glucose units observed. The interaction is defined by two aromatic amino acid/glucose ring interactions between the first glucose unit and Trp 278 and between Tyr 275 and the second glucose unit (Figure 4c). Though only two glucose units are visible in this binding site, it sits between the extended binding site I and binding site IX, lying about 6 Å from the malto-oligosaccharide in binding site I. However, its relative orientation is more consistent with an α-1,6 branched attachment to the glucan in binding site I.

Only a single glucose is visible in the final two sites (sites XI and XII). Site XI is located on the “bottom” of the enzyme, on the side opposite to the active site. It bridges the catalytic domain and the C-terminal domain and is near no other binding site.

Lastly, site XII is located in a region expected to be part of the donor chain binding site. An overlay of the M7-bound *Cyanothece* structure and M8-bound EcBE shows the two sites to be almost in contact (Appendix A). The position of this site may suggest a difference in the trajectory of the donor strand in EcBE after the sixth or seventh residue from the active site; however, with only one glucose visible, it is difficult to make firm conclusions.

## 3. Discussion

The recent structure of M7-bound BE from a cyanobacterial (*Cyanothece*) species has provided important insights into BE structure and function because, for the first time in a BE, it has a donor glucan chain bound in the active site of the enzyme [30] (PDB 5GQX). Many of the residues found to interact with this M7 are conserved in EcBE, as they are with most BEs, indicating a similar trajectory for the donor chain on the surface of BE enzymes (Figure 5 and Appendix A). An overlay of the M7-bound cyanobacterial structure on M8-bound EcBE shows that 12 of the 15 residues found to interact with this M7 are conserved or structurally similar (Figure 5). The three exceptions are W399, V282, and E284. W399 is conserved (corresponding to Trp 336 in EcBE) but resides on a disordered loop in all EcBE structures determined so far. Interestingly, though Trp399 is conserved in the *Mycobacterium tumefaciens* sequence, the structure of the loop it resides on is completely different in the two structures (Appendix A). The significance of this difference is unclear at this point.

Val282 and Glu284 reside on the loop connecting strand 1 and helix 1 of the catalytic domain and interact with the non-reducing end of M7 in the cyanobacterial structure. This loop is 21 residues shorter in EcBE (amino acids 251–256 versus 264–289 in cyanobacterial BE) and lies relatively far from an M7 modeled into EcBE, leaving the non-reducing end unencumbered (Figure 6a,b). When a glucan is modeled into site VII, a binding site that is only occupied in cyclodextrin-bound EcBE [44], it becomes clear that this glucan binding site is quite close to the non-reducing end of the active site-bound M7 donor strand (Figure 6c), and a clear path is available for a longer glucan to connect the reducing end of an oligosaccharide bound in site I to the non-reducing end of the donor strand (modeled based on the M7-bound *Cyanothece* BE structure). To illustrate the point, a maltotriose was modeled to connect the reducing end of M8 in site I to the non-reducing end of the active site-bound donor strand based on the *Cyanothece* structure (Appendix A). No collisions between protein and oligosaccharide were observed in this model. As previously mentioned, M8 bound in site I snakes around loop 251–256, which makes up most of site VII. Remembering that sites I and VII are conserved in only *enterobacterial* enzymes and the preference of these enzymes for the transfer of longer glucans, it is reasonable to hypothesize that binding from the active site to sites I and VII would provide a binding surface for such longer transferred glucans. This would explain why sites I and VII have significant effects on EcBE activity, even though they are not conserved in other BEs [43]. Together, these results begin to reveal a picture of how various branching enzymes confer their differing transfer chain specificity. The cyanobacterial enzyme, which prefers shorter M6 and M7 chains, has a loop that directly interacts with the non-reducing end of M7, providing a preference for glucans of this length (Figure 6d). On the other hand, this loop is much shorter in enterobacterial enzymes, allowing a glucan to continue from the active site to other disparate binding sites on the surface of the enzyme. These observations suggest that the loop connecting strand 1 and helix 1 (the 251–256 loop in EcBE) may play an important role in determining donor chain length in prokaryotic BEs, controlling the branch chain length and, therefore, the metabolic availability of the storage glycogen in these organisms.

Strikingly, a recent report describing M12-bound rice BE1 showed that the equivalent loop connecting strand 1 and helix 1 in the catalytic domain (residues 189–200 in rice BE1) also plays an essential role in branch chain length specificity [28]. In fact, it was shown that substitution of this and a second loop with that of the highly homologous isoform of rice BE1, rice BEIIb, switched the branch chain specificity between the two from a preference for longer chains (rice BEI) to a relatively strict specificity for chains 6–7 residues in length (rice BEIIb) [28]. It would thus appear that despite the relatively low sequence homology between eukaryotic and prokaryotic enzymes and the very different glucan binding seen in these enzymes, the same part of the structure is responsible for transfer chain length specificity.

In contrast to the apparent similarities in donor strand binding, acceptor strand binding appears to be not so well conserved. This insight is derived from comparisons of bacterial BE structures to the structure of M12-bound rice BE. As previously described, M12 binding in rice BE1 is hypothesized to mimic acceptor strand binding in plant BEs [28]. A comparison of this binding site to that of EcBE shows little structural homology between amino acid residues interacting with M12 and the structures. Furthermore, none of the glucan binding sites identified in any of these structures overlap with the acceptor strand binding identified in rice BE, indicating distinctly different interactions of the acceptor strand in these enzymes. Indeed, acceptor strand binding remains a mystery in bacterial BEs.

## 4. Materials and Methods

### 4.1. Materials

EcBE was overexpressed in *E. coli* and purified, as previously described [25,43,44]. (M8 was produced using the same procedure previously used to produce maltododecaose (M12)) [28]. The purity of M8 was measured to be greater than 90%.

### 4.2. Crystallization and Data Collection

The purified protein was buffer exchanged into 1.3 M ammonium tartrate at pH 7.6 and concentrated to 5 mg/mL. Both the buffer exchange and concentration were performed using Centricon (EMD Millipore) concentrators. Crystals were grown by hanging drop vapor diffusion using 24-well Linbro plates with a 1 mL reservoir volume. For each well, 2 μL of protein solution was added to 2 μL of well solution. Small crystals were observed within two weeks. These crystals were then used for microseeding by hanging drop vapor diffusion over the same reservoir solution to grow crystals suitable for X-ray diffraction. The crystals were observed using well solutions containing 1.3 M ammonium tartrate at pH 7.6 with 10% PEG 4000. The crystals were then soaked in 150 mM maltooctaose for 5.5 h. After soaking, the crystals were transferred to a cryogenic protectant containing the mother liquor and 10% glycerol and flash frozen in liquid nitrogen. Diffraction data were collected at the Advanced Photon Source (APS) (Argonne, DuPage, IL, USA) LS-CAT, (sector 21-ID-D) at 1.00 Å wavelength radiation and 100 K, using a MAR300 detector. Data reduction and scaling were performed using the HKL2000 program package, and the data were further refined using CCP4. The structure was solved by molecular replacement (PDB ID: 1M7X). Multiple cycles of refinement were implemented. Placement of the oligosaccharide chains was performed using COOT (0.8.9.1), and multiple cycles of oligosaccharide placement, rebuilding, and refinement were conducted to achieve the final structure. The final model contained four chains of EcBE, with the fourth chain missing the entire CBM48 domain and having much poorer overall electron density. Data collection and refinement statistics are tabulated in Appendix A.

## 5. Conclusions

The structure of M8-bound EcBE identified that the five new binding sites on the surface of the enzyme provided a more complete picture of malto-oligosaccharide binding in site I and suggested that site I is part of the donor chain binding site for longer donor chains. Together, the results identify the loop responsible for donor chain length specificity in EcBE and suggest that it is the same loop that determines chain length in most, if not all, BEs.

## Figures and Tables

**Figure 1 molecules-28-04377-f001:**
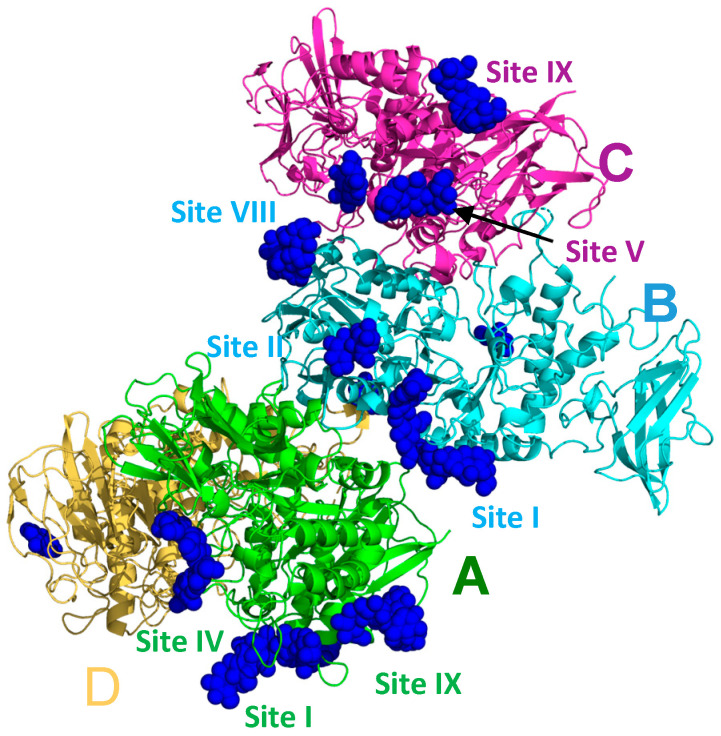
The asymmetric unit of the M8-bound EcBE structure. The four molecules are labeled and colored: (**A**), green, (**B**), cyan, (**C**), magenta, and (**D**), yellow. Bound glucans are shown as blue spheres. A few of the visible glucan-binding sites are labeled, with the label colored by the molecule they are associated with.

**Figure 2 molecules-28-04377-f002:**
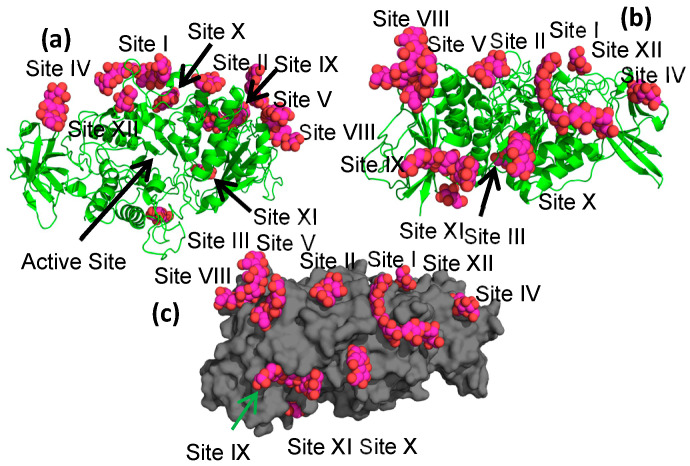
M8-bound EcBE composite structure. (**a**) View down the active site. Made by overlaying the four molecules in the asymmetric unit and showing the most well-ordered or longest glucan for each site. EcBE molecule A is shown as a green cartoon. Glucans are shown as spheres, with C, magenta and O, red. (**b**) Same as part A, but rotated 90° horizontally. (**c**) Same as B, but EcBE shown as a gray surface.

**Figure 3 molecules-28-04377-f003:**
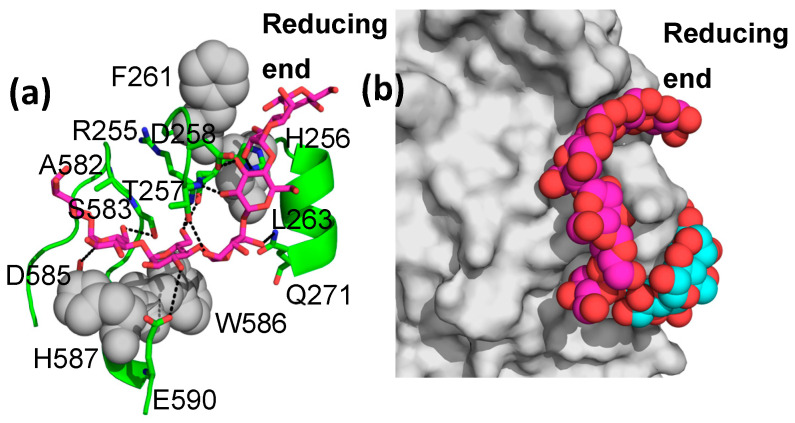
M8 binding in binding site I. (**a**) EcBE is shown as a green cartoon. Amino acids that hydrogen bond with M8 are shown as sticks and colored by atom: C, green, N, blue, and O, red. Residues that make hydrophobic interactions are shown as grey spheres M8 is shown as a stick, with C, magenta, and other atoms colored as above. (**b**) M8 in binding site I showing the alpha-cyclodextrin (colored by atom, C, cyan) modeled into site VII based on the alpha-cyclodextrin-bound EcBE structure (PDBID 5E6Y).

**Figure 4 molecules-28-04377-f004:**
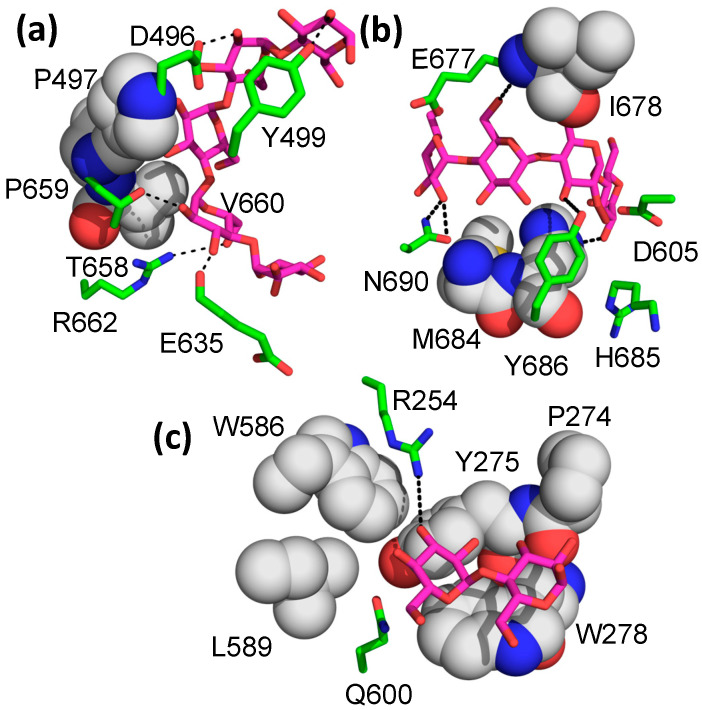
M8 binding in binding sites VIII-X. EcBE M8 interacting residues that make hydrogen bonds are shown as sticks and colored by atom, as previously noted. Residues that make hydrophobic interactions are shown as spheres, with C, gray, and all other atoms as previously noted. (**a**) M8 binding in site VIII. (**b**) M8 binding in site IX. (**c**) M8 binding in site X.

**Figure 5 molecules-28-04377-f005:**
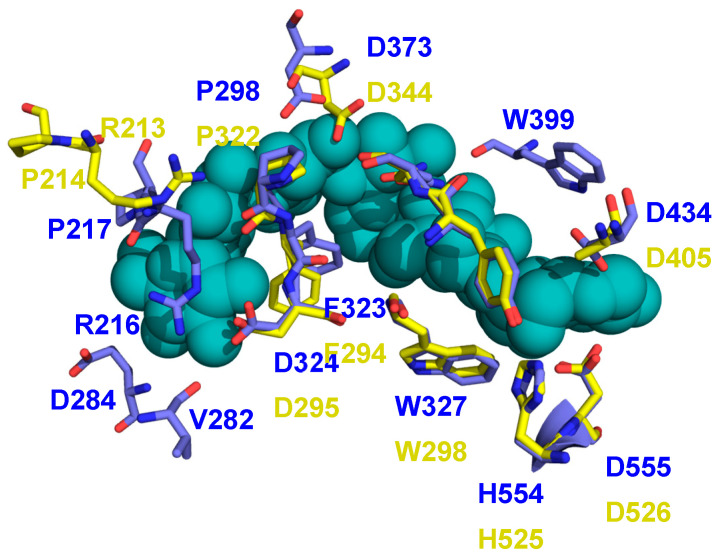
The structures of M8-bound EcBE and M7-bound *Cyanothece* BE are overlayed. Shown is the M7 donor chain bound in the active site of the *Cyanothece* BE (shown as blue–green spheres), with residues in the *Cyanothece* structure that interact shown (C, blue, all other atoms as above) with the equivalent residues in EcBE (C, yellow).

**Figure 6 molecules-28-04377-f006:**
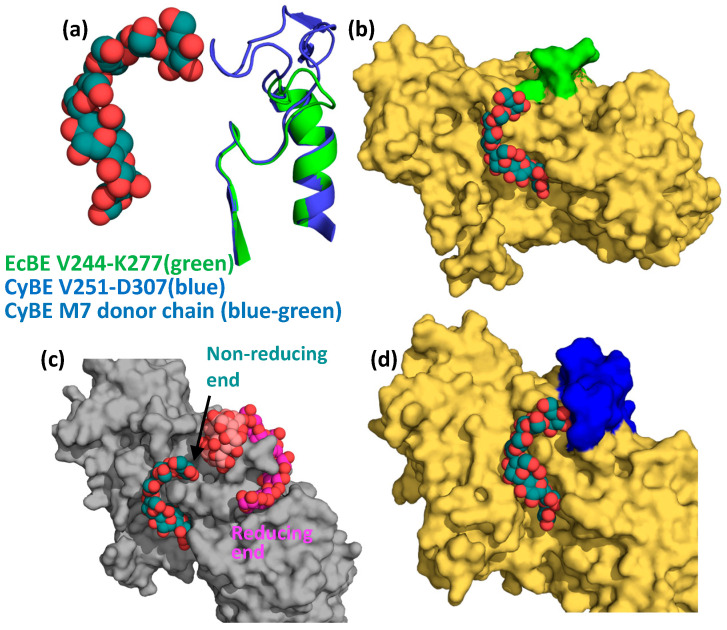
The chain transfer selectivity loop. (**a**) An overlay of M7-bound *Cyanothece* BE (blue cartoon, active site bound M7, spheres, with C, blue–green) and EcBE (green cartoon), showing the V244-K277 loop (EcBE numbering) of each. (**b**) Active site-bound M7 from *Cyanothece* BE is modeled into the EcBE structure (represented as a surface in yellow), with the V244-K277 loop colored green. (**c**) M8-bound EcBE structure showing site I-bound M8 (spheres, C, pink) with alpha-CD modeled into site VII (based on the alpha-CD bound EcBE structure, spheres, C, light red) and active site-bound M7 from *Cyanothece* BE modeled into the active site (spheres, C, blue–green). (**d)** Active-site M7-bound *Cyanothece* BE structure, represented as a yellow surface with V251-D307 loop colored dark blue. M7 colored as above.

**Table 1 molecules-28-04377-t001:** The 12 glucan binding sites identified in glucan-bound EcBE structures.

Site	Residue	Molecules in A.S. ^1^	Number of Glucoses
I	W586	A, B	5, 7
II	W595	A, B, C	1, 2, 2
III	W478	B	1
IV	W159	A	3
V	W544	A, C	1, 3
VI	W628	None	0 ^2^
VII	W262	None	0 ^2^
VIII	Y499	B	5
IX	H685	A, C	5, 4
X	W278	A	2
XI	H613	B	1
XII	F294	A	1

^1^ A.S.: Asymmetric unit. Molecules in the A.S. are labeled A–D. ^2^ These sites were identified in M7-bound EcBE and cyclodextrin-bound EcBE structures.

## Data Availability

Crystallographic structure factors and coordinates were deposited in the Protein Data Bank https://www.rcsb.org/ with PDBID: 8SDB.

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
