# Peer review of "The Structure of Maltooctaose-Bound Escherichia coli Branching Enzyme Suggests a Mechanism for Donor Chain Specificity"

_molecules, 2023, doi:10.3390/molecules28114377_

Round 1

Reviewer 1 Report

The manuscrpt compares the structure of Escherichia coli glycogen branching enzyme binding with maltooctaose with other enzymes.

By the way, it is very difficult to review this paper because there are to many basic problems in descriptions.

1. title: E. coli must be Italic. Use maltooctaose instead malto-octaose. Don't use upper cases improperly.

2. abstract: why did you send 2/3 of abstact for introduction?   It's strage structure of abstract. You failed to show detailed finding in your research. line 14:, it's? its?

3. Keywords: upper cases? remove "glycosyl hydrolase or GH13"

4. line 32, what is the upper case 1?

5. You should put the reference to proper position. And period should be after the reference number.

6. line 61, why do you use improper upper cases?

7. Figure quality is too low.

8. Method sections are too simple. Is that enough? Method description is not formal. Must be improved.

There are too many typos. It should be improved properly.

Author Response

The manuscrpt compares the structure of Escherichia coli glycogen branching enzyme binding with maltooctaose with other enzymes.

By the way, it is very difficult to review this paper because there are to many basic problems in descriptions.

  1. title: E. coli must be Italic. Use maltooctaose instead malto-octaose. Don't use upper cases improperly.

We thank the reviewer for noticing this and it has been corrected.

  1. abstract: why did you send 2/3 of abstact for introduction?   It's strage structure of abstract. You failed to show detailed finding in your research. line 14:, it's? its?

We thank the reviewer for this observation, and we agree and have completely rewritten the abstract to better reflect the conclusions in the paper.

  1. Keywords: upper cases? remove "glycosyl hydrolase or GH13"

GH13 was removed from the keyword list.

  1. line 32, what is the upper case 1?

The superscript 1 was removed.

  1. You should put the reference to proper position. And period should be after the reference number.

We thank the reviewer for noticing this oversight. We have corrected the placement of the reference number throughout the text.

  1. line 61, why do you use improper upper cases?

We have fixed the uppercase error.

  1. Figure quality is too low.

We thank the reviewer for pointing this very important issue out. However, the figure quality in the word document we have is of quite high quality. We will make sure that only high resolution, and high quality figures will be in the manuscript.

  1. Method sections are too simple. Is that enough? Method description is not formal. Must be improved.

We have rewritten the methods section. However, we have not included the methods that have been previously published, which leaves the production of the new crystal form, M8 soaking, data collection and refinement are the only new experimental details germane to this publication.

Reviewer 2 Report

Authors report here the crystal structure of a malto-octaose-bound branching enzyme of the GH13 family from the Enterobacteria E. coli. They identified at the surface of the 3.1 Angstom resolution structure three new binding sites. By comparison with existing structures they propose a new mechanism to explain transfer chain specificity involving some of these surface binding sites.

Overall, the work is very interesting and well explained and presented. I have only minor concerns.

Main document:

Overall, the images resolution is very low but it is probably due to the review process and quality will be improved later on?

Line 74: C. glabrata instead of C. galbrata.

Figure 3: As the orientation is different between Fig3a and 3b, it would be very helpful to indicate the reducing and non-reducing ends on the linear oligosaccharide.

Legend Figure 3: C, green, n N, blue, O, red.

Line 190: Figure 4b B.

Line 196: Figure 4c C.

Line 193: it is stated that in site X, “only two glucose units observed”. However, in figure 4c, three glucose units are visible. Could you explain?

Figure S3 (line 205) is before Figure S2 (line 225).

Legend Figure 5: Line 231, EcBE instead of EcBebE I guess?

Line 239-240: It is stated “a clear path is available for a longer glucan to connect the reducing end of an oligosaccharide bound in site I to the non-reducing end of the hypothesized donor strand”. It is not absolutely clear to me; would it be possible to illustrate the phenomenon by indicating in figure 6 the reducing and non-reducing ends as well as the “hypothesized donor strand”.

Supplementary:

The “Table of contents” does not correspond to the supplementary information.

Legend Figure S2: “with the cyclodextrin bound in site xxx ?? from the alpha-CD-bound EcBE structure (light red C atoms) and the Cyanothece M7 donor chain modeled onto the structure (which color ??). 

Table SI: The resolution is 3.0 Angstrom in the table but 3.1 in the main text.

Author Response

Reviewer 2

Open Review

Quality of English Language

( ) I am not qualified to assess the quality of English in this paper
( ) English very difficult to understand/incomprehensible
( ) Extensive editing of English language required
( ) Moderate editing of English language
( ) Minor editing of English language required
(x) English language fine. No issues detected

Yes

Can be improved

Must be improved

Not applicable

Does the introduction provide sufficient background and include all relevant references?

(x)

( )

( )

( )

Are all the cited references relevant to the research?

(x)

( )

( )

( )

Is the research design appropriate?

(x)

( )

( )

( )

Are the methods adequately described?

(x)

( )

( )

( )

Are the results clearly presented?

( )

(x)

( )

( )

Are the conclusions supported by the results?

(x)

( )

( )

( )

Comments and Suggestions for Authors

Authors report here the crystal structure of a malto-octaose-bound branching enzyme of the GH13 family from the Enterobacteria E. coli. They identified at the surface of the 3.1 Angstom resolution structure three new binding sites. By comparison with existing structures they propose a new mechanism to explain transfer chain specificity involving some of these surface binding sites.

Overall, the work is very interesting and well explained and presented. I have only minor concerns.

Main document:

Overall, the images resolution is very low but it is probably due to the review process and quality will be improved later on?

 We appreciate the reviewer noticing this as it is a very important concern. The resolution and quality of the figures in our original word document is quite high. We conclude that something went wrong during the distribution to the reviewers. We will make sure that the quality of the published figures is of high quality.

Line 74: C. glabrata instead of C. galbrata.

 We have corrected this error.

Figure 3: As the orientation is different between Fig3a and 3b, it would be very helpful to indicate the reducing and non-reducing ends on the linear oligosaccharide.

 We thank the reviewer for this important observation. We have labeled the reducing end of the maltooligosaccharide in both parts of the figure.

Legend Figure 3: C, green, n N, blue, O, red.

We thank the reviewer for noticing this mistake. It has been corrected.

Line 190: Figure 4b B.

Line 196: Figure 4c C.

 Both of these errors have been corrected.

Line 193: it is stated that in site X, “only two glucose units observed”. However, in figure 4c, three glucose units are visible. Could you explain?

We very much thank the reviewer for pointing this out. We have conducted multiple refinements on this structure and have attempted to build glucose units into the binding sites to the best of our ability. Though we had three glucose units in site X for most of the refinement process, we decided to only include the two best ordered in the final structure. Unfortunately, Figure 4c was not updated to reflect this. It has now been corrected to show only the two glucose units we have the best density for.

Figure S3 (line 205) is before Figure S2 (line 225).

 We have changed to order of the figures in the SI and corrected the figure names as appropriate.

Legend Figure 5: Line 231, EcBE instead of EcBebE I guess?

 We have fixed this mistake.

Line 239-240: It is stated “a clear path is available for a longer glucan to connect the reducing end of an oligosaccharide bound in site I to the non-reducing end of the hypothesized donor strand”. It is not absolutely clear to me; would it be possible to illustrate the phenomenon by indicating in figure 6 the reducing and non-reducing ends as well as the “hypothesized donor strand”.

 We thank the reviewer for this observation. We have now labeled Figure 6c to show the non-reducing end of the donor strand (based on the M7-bound Cyanothece structure) and the reducing end of M8 bound in site I. We have reworded the section to better clarify that by “hypothesized donor strand” we meant the M7 based on the Cyanothece structure. We have also included an additional figure (Figure S4) that shows a maltotriose modeled to connect the two strands as we predict, showing the clear pathway and that no protein collisions are required to connect the two oligosaccharides.

Supplementary:

The “Table of contents” does not correspond to the supplementary information.

 The table of contents for the SI has been fixed.

Legend Figure S2: “with the cyclodextrin bound in site xxx ?? from the alpha-CD-bound EcBE structure (light red C atoms) and the Cyanothece M7 donor chain modeled onto the structure (which color ??). 

We have added the color (blue) to the figure caption and have replaced xxx with site VII.

Table SI: The resolution is 3.0 Angstrom in the table but 3.1 in the main text.

We have corrected this in the main text.

Reviewer 3 Report

The manuscript builds on previous work that provided the crystal structure of an E. coli branching enzyme bound to cyclodextrins and linear malto-oligosaccharides to document the crystal structure of the same enzyme with maltoheptaose. Interpretation of the crystal structure documented 5 additional binding sites of the E. coli BE with maltooctaose relative to maltoheptaose, and implied that the enzyme also binds longer donor molecules, corresponding to a less densely packed glycan with longer branch length. The results are appropriately discussed in the context of the ecology of E. coli as a commensal member of intestinal microbiota. Overall, the manuscript is well written; minor suggestions for improvement are indicated below.

Specific comments:

Title and throughout. E. coli should be written in full (Escherichia coli) in the title and on first mention in the text. Latin taxonomic names of organisms should be italicized also in the list of references.

line 42. E. coli predominantly but not exclusively lives in vertebrate hosts – several lineages of the species as well as closely related species, e.g. Escherichia fergusonii, are associated with environmental establishment niches. Lineages that are predominantly found in vertebrate intestines maintain the ability to persist for a long time in extra-intestinal and low-nutrient persistence niches.

line 74 and throughout. Names of genera should be written in full and in capital letters on first mention; species names are italicized but not capitalized.

line 74. Candida glabrata.

line 87. On E. coli as being “exclusive” to vertebrate intestines, see above.

line 224. Mycobacterium tumefaciens;

line 243. “enterobacterial” is an English word and not italicized.

line 287. Here, the source and purity of maltooctaose should be specified. Most commercial preparations have a purity of only 60%, i.e. they contain a substantial proportion of oligosaccharides with shorter or longer degree of polymerization.

line 314. In addition to their role in the ecology of microbes and eukaryotes, branching enzymes are used commercially for starch modifications. The authors may or may not wish to comment on how data on the donor length specificity impact on this use of the enzyme.

Other than nomenclature edits, the quality of English does not need to be improved.

Author Response

Reviewer 3

Open Review

Quality of English Language

( ) I am not qualified to assess the quality of English in this paper
( ) English very difficult to understand/incomprehensible
( ) Extensive editing of English language required
( ) Moderate editing of English language
(x) Minor editing of English language required
( ) English language fine. No issues detected

Yes

Can be improved

Must be improved

Not applicable

Does the introduction provide sufficient background and include all relevant references?

(x)

( )

( )

( )

Are all the cited references relevant to the research?

(x)

( )

( )

( )

Is the research design appropriate?

(x)

( )

( )

( )

Are the methods adequately described?

( )

(x)

( )

( )

Are the results clearly presented?

(x)

( )

( )

( )

Are the conclusions supported by the results?

(x)

( )

( )

( )

Comments and Suggestions for Authors

The manuscript builds on previous work that provided the crystal structure of an E. coli branching enzyme bound to cyclodextrins and linear malto-oligosaccharides to document the crystal structure of the same enzyme with maltoheptaose. Interpretation of the crystal structure documented 5 additional binding sites of the E. coli BE with maltooctaose relative to maltoheptaose, and implied that the enzyme also binds longer donor molecules, corresponding to a less densely packed glycan with longer branch length. The results are appropriately discussed in the context of the ecology of E. coli as a commensal member of intestinal microbiota. Overall, the manuscript is well written; minor suggestions for improvement are indicated below.

Specific comments:

Title and throughout. E. coli should be written in full (Escherichia coli) in the title and on first mention in the text. Latin taxonomic names of organisms should be italicized also in the list of references.

We have fixed this in both the title and the rest of the manuscript.

line 42. E. coli predominantly but not exclusively lives in vertebrate hosts – several lineages of the species as well as closely related species, e.g. Escherichia fergusonii, are associated with environmental establishment niches. Lineages that are predominantly found in vertebrate intestines maintain the ability to persist for a long time in extra-intestinal and low-nutrient persistence niches.

We thank the reviewer for this astute observation. We have corrected the text to reflect the points the reviewer has made. We have also included a more detailed description, with additional references, describing the hypothesis that glycogen can act as a durable energy reserve in some procaryotic organisms (page 2, middle paragraph).

line 74 and throughout. Names of genera should be written in full and in capital letters on first mention; species names are italicized but not capitalized.

We have corrected this issue throughout the text.

line 74. Candida glabrata.

We have corrected the spelling and capitalization.

line 87. On E. coli as being “exclusive” to vertebrate intestines, see above.

We have modified the text.

line 224. Mycobacterium tumefaciens;

We removed the capitalization.

line 243. “enterobacterial” is an English word and not italicized.

We fixed this error.

line 287. Here, the source and purity of maltooctaose should be specified. Most commercial preparations have a purity of only 60%, i.e. they contain a substantial proportion of oligosaccharides with shorter or longer degree of polymerization.

Our M8 was produced as previously described for M12 by our co-author. A description of the protocol is given in a previous reference. We have modified the experimental to reflect this.

line 314. In addition to their role in the ecology of microbes and eukaryotes, branching enzymes are used commercially for starch modifications. The authors may or may not wish to comment on how data on the donor length specificity impact on this use of the enzyme.

 We have included a brief mention of BE’s commercial application in the introduction (page 2).

Round 2

Reviewer 1 Report

The authors present interesting findings about the binding of maltooctaose and N-terminal domain truncated Escherichia coli branching enzyme. The research provides new insight on the effects of sugars binding on the structural and functional properties of EcBE.

By the way, there are many small errors and typos that might be easily corrected. 

1.   Abstract: line 22-24, the expression is not clear. Clarify the meaning

2. Abstract: line 25, "from all organisms" is too broad expression.

3. Abstract: do not use upper case for branching enzyme used in the middle of sentences.

4. line 127, what do you mean? "it's metabolism"?

5. line 157, when did you access the website?

6. Pay more attention on the names of microorganisms

7. line 175, do not use this kind of abbreviated expression.

8. Fig 1. provide binding site numbers that are shown in Table 1 on the figure.

There are still many minor problems of expressions and format. Please pay more attention on it.

Author Response

We would like to thank the reviewer for their suggestions, which further improve the quality of our manuscript. We document the changes we’ve made (in italics) in response to the reviewers comments.

The authors present interesting findings about the binding of maltooctaose and N-terminal domain truncated Escherichia coli branching enzyme. The research provides new insight on the effects of sugars binding on the structural and functional properties of EcBE.

By the way, there are many small errors and typos that might be easily corrected. 

We have read through the manuscript again carefully, and have attemped to correct any other mistakes.

  1. Abstract: line 22-24, the expression is not clear. Clarify the meaning

We have rewritten the sentence to improve clarity.

  1. Abstract: line 25, "from all organisms" is too broad expression.

We have altered the text to “a diversity of organisms”

  1. Abstract: do not use upper case for branching enzyme used in the middle of sentences.

We have fixed this.

  1. line 127, what do you mean? "it's metabolism"?

We’ve corrected this.

  1. line 157, when did you access the website?

We apologize, but we do not understand the meaning of this comment. We frequently access the CAZY website for reference. At least form the time that we have accessed the website, the statements made here have been demonstrably correct.

  1. Pay more attention on the names of microorganisms

We have checked and corrected the names of microorganisms.

  1. line 175, do not use this kind of abbreviated expression.

We have corrected this.

  1. Fig 1. provide binding site numbers that are shown in Table 1 on the figure.

We have labeled some of the binding sites on figure 1, but since many of the binding sites are not visible in a single orientation, attempting to label all 15 of them would be uninterpretable.

Comments on the Quality of English Language

There are still many minor problems of expressions and format. Please pay more attention on it.